# Suppressing MDSC Recruitment to the Tumor Microenvironment by Antagonizing CXCR2 to Enhance the Efficacy of Immunotherapy

**DOI:** 10.3390/cancers13246293

**Published:** 2021-12-15

**Authors:** Kennady Bullock, Ann Richmond

**Affiliations:** 1Department of Pharmacology, Vanderbilt University School of Medicine, Nashville, TN 37232, USA; kennady.k.bullock@vanderbilt.edu; 2Department of Veterans Affairs, Tennessee Valley Healthcare System, 432 PRB, 2220 Pierce Ave, Nashville, TN 37232, USA

**Keywords:** myeloid-derived suppressor cells (MDSCs), CXCR2, immunotherapy resistance, immune checkpoint inhibitors

## Abstract

**Simple Summary:**

While the development of immunotherapy has greatly advanced cancer treatment, many patients do not benefit from immunotherapy. Numerous strategies have been developed to improve response to immunotherapy across cancer types, including blocking the activity of immunosuppressive immune cells, cytokines, and signaling pathways that are linked to poor responses. Myeloid-derived suppressor cells (MDSCs) are associated with poor responses to immunotherapy, and the chemokine receptor, CXCR2, is involved in recruiting MDSCs to the tumor. In this review, we present studies that explore the potential of inhibiting MDSCs through blocking CXCR2 as a strategy to enhance response to existing and novel immunotherapies.

**Abstract:**

Myeloid-derived suppressor cells (MDSCs) are a heterogenous population of cells derived from immature myeloid cells. These cells are often associated with poor responses to cancer therapy, including immunotherapy, in a variety of tumor types. The C-X-C chemokine receptor 2 (CXCR2) signaling axis plays a key role in the migration of immunosuppressive MDSCs into the tumor microenvironment (TME) and the pre-metastatic niche. MDSCs impede the efficacy of immunotherapy through a variety of mechanisms. Efforts to target MDSCs by blocking CXCR2 is an active area of research as a method for improving existing and novel immunotherapy strategies. As immunotherapies gain approval for a wider array of clinical indications, it will become even more important to understand the efficacy of CXCR2 inhibition in combating immunotherapy resistance at different stages of tumor progression.

## 1. Introduction

According to the concept of cancer immune editing, tumor cells progress through three stages of interaction with the immune system: elimination, equilibrium, and escape [1]. Therefore, clinically diagnosable tumors have evolved to bypass the immune system’s natural, protective mechanisms. Immunotherapy strategies, including immune checkpoint inhibitors, cancer vaccines, adoptive T-cell therapy, and genetically modified immune cells, seek to activate the patient’s immune system against tumor cells that have progressed through the escape phase [2]. FDA-approved immune checkpoint inhibitors include antibodies against the following checkpoint proteins: cytotoxic T-lymphocyte-associated protein 4 (CTLA4), programmed cell death protein 1 (PD1), and programmed death-ligand 1 (PD-L1). Immune checkpoint inhibitors have greatly expanded treatment options for patients with late-stage, metastatic disease [3], and are currently being investigated as treatment strategies in the neoadjuvant setting [4]. Despite long-term survival benefits in some patients, a large patient population remains that either does not respond to immune checkpoint inhibitors or develops resistance after an initial period of response, necessitating the development of treatment strategies that overcome resistance. As immunotherapy expands for the treatment of earlier-stage tumors, additional markers and mechanisms of resistance will likely emerge. Resistance to immunotherapy can occur via tumor cell-extrinsic and/or tumor cell-intrinsic mechanisms [5], and targeting MDSCs through CXCR2 inhibition is an emerging strategy to counteract both mechanisms of resistance.

## 2. The Role of MDSCs in the Establishment of an Immunosuppressive Niche 

Tumor cell-extrinsic mechanisms of immunotherapy resistance include low infiltration of tumor antigen-specific T cells and the presence of immunosuppressive cells such as MDSCs and tumor-associated macrophages (TAMs) [6]. Elevated numbers of MDSCs are associated with poor response to therapy, including, immunotherapy across several tumor types [7,8,9]. High numbers of MDSCs in the peripheral blood were associated with poor overall survival in a phase I/II study of melanoma patients treated with anti-PD1 therapy after progressing on anti-CTLA4 therapy [10]. Similarly, melanoma patients being treated with anti-CTLA4 with lower levels of MDSCs following their first infusion had increased overall survival compared to patients with higher circulating MDSCs [11]. 

Once in the tumor microenvironment (TME), MDSCs contribute to immune suppression through a variety of mechanisms including production of inducible nitric oxide synthase (iNOS), arginase 1 (ARG1), transforming growth factor-beta (TGFβ), IL-10, cyclooxygenase-2 (COX2), and indoleamine 2,3-dioxygenase (IDO) [12]. MDSCs can also suppress CD8+ T-cell activity through the expression of Fas-ligand, which interacts with Fas expressed on tumor-infiltrating lymphocytes (TILs) to induce TIL apoptosis [13]. Blocking this interaction with soluble Fas-Fc restored sensitivity to immune checkpoint blockade therapy in a mouse model of melanoma [13]. Additionally, MDSCs help recruit other immunosuppressive cells such as TAMs. In a murine model of prostate cancer, administration of a CXCR2 antagonist or infusion of bone marrow-derived CXCR2 KO macrophages led to a reduction in tumor growth and a reprogramming of TAMs to a pro-inflammatory, M1 phenotype [14]. As well as promoting immunosuppression, MDSCs are also involved in various aspects of tumorigenesis including epithelial to mesenchymal transition (EMT) [15], angiogenesis [16], and establishment of the pre-metastatic niche [17]. Due to their multifaceted roles in immunosuppression, promotion of tumor growth, and treatment resistance, MDSCs are a desirable therapeutic target.

### MDSC Classifications

There are two main classifications of MDSCs, granulocytic/polymorphonuclear (G-MDSCs/PMN-MDSCs) and monocytic (M-MDSCs), each with distinct surface protein expression profiles. The terms G-MDSCs and PMN-MDSCs may be used interchangeably. Mouse markers of G-MDSCs and M-MDSCs include CD11b^+^Ly6G^+^Ly6C^lo^ and CD11b^+^Ly6G^−^Ly6C^hi^, respectively, while human G-MDSCs can be defined as CDllb^+^CD14^−^CD15^+^CD66^+^Lox-1^+^ and human M-MDSCs are defined as CD14^+^CD15^−^HLA^−^DR^−/l0^ [8]. Both types of MDSCs emerge from common myeloid progenitor (CMP) cells, but M-MDSCs more closely follow the differentiation pathway of monocytes whereas G-MDSCs more closely follow the differentiation pathway of granulocytes [18,19] (Figure 1). In mouse models, it is difficult to distinguish M-MDSCs from monocytes and G-MDSCs from granulocytes due to the lack of unique surface markers and the reliance on functional assays to identify MDSCs. However, CD14 was recently described as a marker to distinguish PMN-MDSCs from classical neutrophils in tumor-bearing mice [20]. In contrast, human M-MDSCs and G-MDSCs are more readily distinguished from monocytes and granulocytes. Human M-MDSCs can be distinguished from monocytes by the expression of HLA-DR [12], and Lectin type oxidized receptor I (LOX-1) can be used to distinguish PMN-MDSCs from neutrophils both in peripheral blood and in tumor tissue [21]. Furthermore, G-MDSCs are lower density cells and can be separated from neutrophils by Ficoll gradient centrifugation of peripheral blood mononuclear cells (PMBCs) [22]. G-MDSCs are also very similar to another cell type—the tumor-associated neutrophil (TAN). TANs can be polarized and have been described as N1 TANs that have anti-tumorigenic activity, and N2 TANs that are pro-tumorigenic [23]. There is much debate surrounding the distinctions, if any, between G-MDSCs and N2 TANs [24]. M-MDSCs can further differentiate into tumor associate macrophages (TAMs) and inflammatory dendritic cells in the tumor microenvironment [25,26,27], while G-MDSCs are a differentiated, pathologically activated cell type [28]. A small population of M-MDSCs was initially postulated to be able to differentiate into G-MDSCs through downregulation of the retinoblastoma 1 (*Rb1*) gene [29]; however, this precursor population was later identified as monocyte-like precursors of granulocytes (MLPGs), and MLPGs contributed to up to 50% of the total G-MDSCs in some mouse tumor models [30]. 

## 3. The Role of CXCR2 in Tumor Cell-Intrinsic Mechanisms of Immunotherapy Resistance

In a physiological setting, CXCR2 is primarily expressed by neutrophils and is essential for the recruitment of leukocytes to sites of inflammation, infection, or tissue damage [31]. In the context of cancer, CXCR2 plays a major role in the recruitment of MDSCs to the TME [32]. CXCR2 is a G protein-coupled receptor (GPCR) that signals through G_i_-coupled mechanisms, and the crystal structure of CXCR2 bound to an antagonist was recently determined [33]. CXCR2 activation can also occur via the formation of heterodimers with the atypical chemokine receptor CCRL2, and this is involved in neutrophil recruitment to sites of inflammation [34]. CXCR1 is a GPCR closely related to CXCR2 that is involved in neutrophil and MDSC chemotaxis in humans; however, a role for mouse CXCR1 involvement in neutrophil or MDSC recruitment has not been established [35], [36]. In humans and mice, CXCR2 is expressed by tumor cells, neutrophils, mast cells, monocytes, macrophages, endothelial cells, muscle cells, and epithelial cells [31]. CXCR2 is also expressed on MDSCs and was identified as part of an MDSC gene signature in a scRNAseq analysis of MDSCs from tumors and spleens of MMTV-PYMT mice [37]. The major CXCR2 ligands associated with MDSC chemotaxis are CXCL5, CXCL2, CXCL1, and CXCL8 (IL-8) [38] (Figure 2). It is important to note, however, that while CXCL8 is a major CXCR2 ligand in humans, it is not expressed in mice [31]. Tumor cell-intrinsic mechanisms of immunotherapy resistance include alterations in tumor cell signaling pathways that alter interactions with immune cells in the TME. Increased tumor cell secretion of CXCR2 ligands can contribute to resistance to immunotherapy. In studies using 3D cell cultures and the 4T1 mouse model of breast cancer, tumor-secreted CXCR1/2 ligands induced the formation of neutrophil extracellular traps (NETs) which block the contact of cytotoxic T cells and NK cells with tumor cells. Inhibition of this process with protein arginase deiminase 4 (PAD4) inhibitors led to an increase in sensitivity to the combination treatment of anti-PD1 + anti-CTLA4 [39]. 

## 4. Expression of CXCR2 and CXCR2 Ligands Is Associated with Poor Response to Therapy 

CXCR2 and CXCR2 ligands are markers of poor prognosis in several tumor types. A meta-analysis of 4012 patients with solid tumors from 21 different studies found that CXCR2 expression was predictive of poor prognosis of patients with hepatocellular carcinoma, gastric cancer, or esophageal cancer [40]. Similarly, CXCR2 expression was identified as a poor prognostic marker in lung cancer [41], pancreatic ductal adenocarcinoma [42], and colorectal cancer [43]. Elevated serum levels of the CXCR2 ligands CXCL1 and CXCL2 correlated with increased intra-tumoral MDSC infiltration and reduced overall survival in a cohort of ovarian cancer patients [44], while elevated serum CXCL8 was associated with metastasis and reduced overall survival in pediatric patients with rhabdomyosarcoma [45]. CXCL8 in particular has been found to associate with poor response to immunotherapy [46,47]. A retrospective analysis of 1344 patients across four phase-3 clinical trials, encompassing patients receiving treatment for melanoma, NSCLC, and renal cell carcinoma found an association between serum CXCL8 levels and poor response to immune checkpoint inhibition [48]. Interestingly, high expression of CXCR2 was associated with higher-grade tumors but longer relapse-free survival and higher TIL infiltration in a cohort of triple-negative breast cancer patients treated with adjuvant chemotherapy [49]. The cell-type specificity of CXCR2 expression is important when considering correlations with clinical outcomes, and they found that CXCR2 expression by immunohistochemistry primarily overlapped with the neutrophil markers CDllb and CD66b [49]. It is possible that these neutrophils were phenotypically N1, anti-tumor and that the presence of these TANs contributed to the longer relapse-free survival observed in this patient cohort. Several other studies identified CXCR2 as a key player in establishing the metastatic niche of breast cancer [14,44,45]. In the 4T1 murine tumor model of triple-negative breast cancer, CXCR2+ MDSCs are elevated in lung and lymph node metastases [15,50]. Bone is one of the most common sites of breast cancer metastasis, and in studies using ex vivo cultures of mouse long bones co-cultured with PyMT tumor cells, CXCR2 drove tumor cell colonization in the bone explants [51]. Furthermore, CXCR2 was expressed on G-MDSCs in the tumor microenvironment in the E0771-luciferase murine model of triple-negative breast cancer [52]. 

## 5. CXCR2 Inhibition May Enhance the Efficacy of Existing Immunotherapy Agents

Given the role that CXCR2 plays in recruiting MDSCs to the tumor microenvironment, CXCR2 inhibition is a promising strategy to relieve MDSC-mediated immunosuppression and improve the effectiveness of existing immunotherapies. As immunotherapies gain approval for a wider variety of cancer types and stages, it will become more important to better understand the particular indications for which CXCR2 inhibition may provide synergistic benefit when paired with immunotherapy. The immunologic milieu of tumors at the primary site differs from tumors at metastatic sites, suggesting that immunomodulatory strategies can have divergent effects on primary versus metastatic tumors [53]. CXCR2 is a well-studied player in the establishment in the metastatic niche in a variety of cancers [54]. In a study of surgical specimens from patients with colorectal cancer, CXCR2 expression was significantly increased in tumors from patients who had distant metastases in the lung or liver compared to patients with localized disease [55]. In a murine model of breast cancer metastasis, lungs from tumor-bearing, myeloid-specific CXCR2 KO mice had a decreased M2 macrophage population and an increased CD8+ T-cell population as compared to WT controls [56]. Furthermore, myeloid-specific CXCR2 KO mice displayed decreased intra-tumor infiltration of MDSCs both in models of breast cancer and melanoma, and those remaining MDSCs were less functional [56]. The role of CXCR2 in promoting metastasis alongside the evidence of elevated CXCR2 in metastatic disease compared to localized disease may suggest that patients with metastatic disease will benefit from a combination of immunotherapy and CXCR2 antagonism. In a murine model of melanoma, inhibiting CXCR2 after surgical resection of the primary tumor significantly extended survival and reduced the incidence of distant metastases [57]. In the spontaneous KPC mouse model of pre-invasive pancreatic cancer, there were no differences in overall or tumor-free survival between wild-type and CXCR2−/− mice; however, genetic deletion of CXCR2 significantly reduced metastasis in mice greater than 10 weeks old [42]. Pharmacological inhibition of CXCR2 similarly reduced metastasis, suggesting CXCR2 inhibition may be more effective in later-stage tumors after surgical resection of the primary tumor [42]. 

### 5.1. CXCR2 Antagonism in Combination with Immune Checkpoint Inhibitors 

Several pre-clinical studies have examined the synergism of CXCR2 antagonism with immune checkpoint inhibitors and other immunotherapy agents. In studies using the MOC1 (murine oral cancer 1) model, single-agent treatment with either the CXCR1/2 antagonist, SX-682, or PD1-mAb had no significant effect on tumor growth or survival, but combination treatment with SX-682 and PD1-mAb significantly reduced tumor growth, improved survival, and caused complete tumor rejection in 20% of mice [58]. In a different study using the MOC1 tumor model, depletion of G-MDSCs with a Ly6G+ mAb in combination with anti-CTLA4 led to tumor rejection in 11/11 mice, whereas anti-CTLA4 treatment led to tumor rejection in 5/11 mice [59], providing further evidence for the efficacy of strategies that target MDSCs to improve immunotherapy responses. In a PD1-mAb-resistant mouse model of pancreatic cancer, pharmacological inhibition of CXCR2 synergized with anti-PD1 therapy to significantly extend survival, while genetic deletion or inhibition of CXCR2 increased T-cell infiltration into the TME. These effects were dependent on CXCR2 expression on Ly6G+ cells, as Ly6G depletion abrogated these effects [42]. However, it is important to consider that a Ly6G antibody will deplete neutrophils as well as G-MDSCs in this experimental design. Combination treatment with SX-682 and anti-PD1 significantly reduced tumor burden compared to both vehicle control and single-agent treatment groups in a murine model of melanoma [56]. Similarly, in a murine model of rhabdomyosarcoma (RMS), treatment with an anti-CXCR2 antibody sensitized tumors to anti-PD1 treatment [45]. In addition to these promising pre-clinical studies, there are ongoing clinical studies examining CXCR2 antagonism in combination with immune checkpoint inhibitors; SX-682 in combination with pembrolizumab (anti-PD1) for the treatment of metastatic melanoma (NCT03161431), SX-682 in combination with nivolumab (anti-PD1) for metastatic pancreatic ductal adenocarcinoma (NCT04477343) (Table 1). 

### 5.2. CXCR2 Antagonism in Combination with Other Immunotherapy Agents 

Relatively few studies have examined the effects of CXCR2 antagonism in combination with immunotherapy agents other than checkpoint inhibitors. In a study using the murine oral cancer 2 (MOC2) model, administration of the CXCR1/2 small-molecule inhibitor, SX-682, inhibited MDSC accumulation in the tumor and enhanced the efficacy of NK cell-based adoptive cell transfer therapy [60]. CXCR2 antagonism was tested in combination with immunotherapy consisting of adenovirus encoded TNF-related apoptosis ligand (TRAIL) plus TLR9 agonist, CpG, oligonucleotide (AdT + CpG) in a murine model of breast cancer. Interestingly, this combination provided benefit for obese animals, but not for lean animals [52]. Obese mice were found to have an increase in FasL+ G-MDSCs, which mediated apoptosis of CD8 T cells, as well as an increase in CXCR2 ligands. The obesity-associated increase in CXCR2 ligands relative to the lean state [61], likely explains why the obese mice were more responsive to CXCR2 inhibitor treatment. This suggests the interesting possibility that patients with higher levels of CXCR2 ligands may respond better to CXCR2 inhibition. In another study investigating the efficacy of dendritic cell vaccines in a murine model of glioma, treatment with a CXCR2 neutralizing antibody reversed the survival benefits seen with the vaccine [62]. The deleterious effects of blocking CXCR2 in this instance, however, were due to the role that CXCR2 plays in driving the migration of the modified dendritic cells to their therapeutic target. Thus, these data suggest that some combinatorial therapies using CXCR2 antagonists will not be beneficial. As the selection of available immunotherapy strategies widens, it will be important to explore whether the benefits of CXCR2 antagonism will overcome resistance to other types of immunotherapies in pre-clinical models. 

## 6. Novel CXCR2-Based Immunotherapy Strategies

Blocking CXCR2 with small-molecule antagonists is one strategy to enhance the efficacy of immunotherapy for clinical benefit; hijacking the functions of CXCR2 by driving effector immune cells to express CXCR2 is another. The production of CXCR2 ligands in the TME drives the recruitment of CXCR2-expressing cells into the tumor [63]. This finding has important implications for the development of novel immunotherapy strategies. CXCR2-modified chimeric antigen receptor T cells (CAR T cells) are under development to enhance intra-tumoral T-cell infiltration [64,65,66]. CAR T-cell therapy has found its greatest successes in the treatment of hematological malignancies, but successful intra-tumoral infiltration of CAR T cells is the major limiting factor prohibiting the advancement of CAR T-cell therapies in solid malignancies [66]. In a murine model of hepatocellular carcinoma, treatment with CXCR2-modified CAR T cells significantly reduced tumor burden and enhanced intra-tumoral T-cell infiltration [65]. In mouse xenograft models of ovarian and pancreatic cancer, administration of CAR T cells engineered to co-express αvβ6 integrin and CXCR2 significantly reduced tumor growth [64]. In addition to these promising pre-clinical studies, CXCR2-transduced autologous tumor-infiltrating lymphocytes are under clinical investigation in patients with metastatic melanoma (Table 1). Additionally, there are ongoing efforts to target CXCR2+ positive cells using CXCL5-modified nanoparticles to deliver drug [67]. Natural killer (NK) cells are another effector immune cell type that can have direct cytotoxic activities against tumor cells [68]. Primary NK cells from blood and tumor biopsies from patients with renal cell carcinoma engineered ex vivo to express CXCR2, showed enhanced migratory capabilities in vitro [69]. Driving effector cells such as NK cells and T cells to express CXCR2 is a novel strategy for increasing the infiltration of anti-tumor immune cells to the TME. 

## 7. Clinical Status and Concerns of CXCR2 Antagonists

CXCR2 antagonists are under investigation as anti-inflammatory therapeutics in a variety of disease states including chronic obstructive pulmonary disorder (COPD), type I diabetes, rheumatoid arthritis, ulcerative colitis, and cancer [70]. The CXCR1/2 inhibitor, reparixin, showed promise in a window-of-opportunity clinical trial for HER-2-negative breast cancer. Reparaxin was safe, well-tolerated, and caused a reduction in cancer stem cells in patient tumors [71]. A phase 1b trial in patients with metastatic breast cancer found reparaxin to be safe in combination with paclitaxel [72]. When reparaxin was investigated in combination with paclitaxel as a frontline treatment for metastatic triple-negative breast cancer in a phase 2 trial; however, the primary endpoint of prolonged progression-free survival was not met [73]. While there are no CXCR2-targeted drugs currently approved for use in cancer treatment, other CXCR2 antagonists are in various stages of clinical development (Table 1). 

Clinical concerns with targeting CXCR2 include unwanted effects on other cell types as well as the development of resistance mechanisms. The goal of CXCR2 antagonism may be to target immunosuppressive cells in the TME, however expression of CXCR2 is not limited to MDSCSs. CXCR2 is expressed by a variety of other cell types including endothelial cells, epithelial cells, macrophages, and mast cells [31], and the potential off-target effects on these other cell types necessitate further study. The impact of CXCR2 inhibition on the innate effector functions of neutrophils is a well-studied side effect. A phase II clinical study of the CXCR2 antagonist, SCH527123, was discontinued due to neutropenia in healthy subjects [74]. A second CXCR2 antagonist, AZD5069, was studied in patients with COPD. Findings were more promising in that absolute blood neutrophil counts were reduced in the first 24 h of treatment, but this was reversible upon treatment cessation [75]. Moreover, in studies examining the effects of CXCR2 antagonists on the function of neutrophils in blood from healthy patients, it was found that despite transient decreases in neutrophil numbers, key antimicrobial functions such as phagocytosis and oxidative burst were unaffected [76]. Similarly, CXCR2 inhibition did not affect B-cell responses or cell-mediated immunity [77]. These studies provide evidence to support the safety of CXCR2 antagonism, as transient decreases in total neutrophil counts may occur, but the functionality of the neutrophils is unaffected. 

It is important to consider that inhibition of neutrophil recruitment may have both positive and negative effects in the context of cancer treatment. Neutrophils are recruited to sites of inflammation, and there is evidence to suggest that neutrophils are capable of assuming anti-tumor N1 properties, particularly in the instance of early-stage disease [78,79]. In other instances, neutrophils can exhibit N2 phenotypes and promote tumor growth [23]. Recruitment of both N1 and N2 TANs will likely be altered with CXCR2 antagonism. A high neutrophil-to-lymphocyte ratio in peripheral blood correlates with poor overall survival across several solid tumor types [80]. However, neutrophil content in the peripheral blood may not be reflective of neutrophils in the TME. TANs isolated from early-stage human lung tumors exhibited an immune-activated phenotype and stimulated T-cell responses in vitro, suggesting that early-stage TANs possess anti-tumor functions [81]. Neutrophil recruitment to the lung prevented the establishment of lung metastases in a NOD/SCID mouse model using human renal cell carcinoma (RCC) cell lines [82]. Furthermore, poorly metastatic human RCC cell lines exhibited increased secretion of the neutrophil recruiting chemokines, CXCL5 and IL-8, compared to highly metastatic cell lines. Although there are conflicting results regarding the pro- and anti-tumor functions of neutrophils, this may be largely due to discrepancies in nomenclature and inconsistencies in distinguishing G-MDSCs from N2 TANs. Blocking CXCR2 in early-stage disease may prevent the recruitment of anti-tumor neutrophils, but the effect of CXCR2 inhibition on TAN phenotypes necessitates further study and may vary depending upon the proportion of N1 versus N2 TANs in the TME. 

Another clinical concern with blocking CXCR2 is the development of resistance mechanisms such as a compensatory increase in CXCR2 ligands which may cause an aggerated inflammatory response. Breast cancer cell lines treated with the CXCR2 antagonist exhibited an increase in CXCL2 as a resistance mechanism [83]. In a follow-up study, it was found that treatment with a CXCR2 antagonist also caused an increase in CXCL1, but that this rebound increase in CXCR2 ligands could be prevented by treatment with PKC agonists including bryostatin I, FR236924, or Roy-bz [84]. However, this is not yet a viable solution to treatment resistance, as there currently are no clinically approved PKC agonists. Compensatory mechanisms in response to CXCR2 blockade were further studied in a CXCR2 knockout (KO) mouse model. Unexpectedly, KO mice displayed an exaggerated inflammatory response, accompanied by significantly higher levels of CXCL1 in the skin following application of the inflammatory stimulus, tetradecanoyl phorbol 13-acetate (TPA) [85]. The studies using CXCR2 KO mice emphasize the importance of potential compensatory inflammatory pathways that may be activated when CXCR2 is targeted pharmacologically. This is especially important to consider when combining CXCR2 antagonism with immunotherapy as immune-related toxicities and hyperinflammatory responses are frequent reasons for the discontinuation of immunotherapies in patients [86].

## 8. Additional Strategies to Target MDSC Recruitment

Additional pathways beyond the CXCR2 signaling axis are involved in the recruitment of MDSCs and can be targeted pharmacologically in combination with CXCR2 antagonists and/or immunotherapy. One such axis involved in the recruitment of MDSCs is the colony-stimulating factor-1 (CSF1)/colony-stimulating factor-1 receptor (CSF-1R) axis [87]. In a murine orthotopic model of lung cancer, mice treated with a CSF1R inhibitor exhibited increased intra-tumoral G-MDSCs, and this was found to be caused by increased secretion of CXCL1 by cancer-associated fibroblasts (CAFs). Combined treatment with a CSF1R inhibitor and a CXCR2 antagonist led to a significant reduction in tumor growth that was further enhanced by the addition of anti-PD1 [88]. In a murine model of melanoma, CSF-1R blockade successfully depleted MDSCs and also re-sensitized the tumors to therapies targeting CTLA-4, PD-1, and IDO [89]. Another study by Qin et al. sought a novel approach to depleting MDSCs. They used a peptide phage-display library to identify peptide ligands capable of binding to MDSCs and to design a peptibody, an MDSC-specific peptide bound to mouse IgG2b. Treatment with the peptibody caused anti-tumor effects in murine models, and effectively depleted both granulocytic and monocytic MDSCs by recognizing S100 family proteins on the surface of MDSCS [90].

Other chemokine receptors involved in the recruitment of MDSCs to the TME include CCR2, CCR5, CCR1, and CXCR5 [91]. CCR2 blockade to prevent the recruitment of TAMs to the TME has been explored clinically [92]. However, it was found that CCR2 inhibitor treatment can cause a compensatory increase in CXCR2+ neutrophils, leading to treatment failure [93]. Investigating this effect further in a mouse model of pancreatic ductal adenocarcinoma (PDAC), Nywening et al. found that combining CCR2 inhibition with CXCR2 inhibition led to a reduction in tumor growth and an increase in tumor-infiltrating lymphocytes [93]. Single-agent treatment strategies to block the recruitment of immunosuppressive myeloid cells to the TME have failed to find success due to compensatory pathways of myeloid cell recruitment. However, combining CXCR2 inhibition with other strategies to block MDSC recruitment as well as with immunotherapies may prove more successful. 

## 9. Conclusions and Future Directions

The development of immune checkpoint inhibitors and other immunotherapies represents a major advancement in cancer research that has greatly benefited certain patient populations. Nevertheless, many patients either do not respond to immunotherapy or develop resistance after an initially promising response. MDSCs are associated with adverse patient outcomes such as reduced overall survival and poor response to therapy, including immunotherapy [8,10,11]. Targeting MDSC recruitment to the TME by blocking CXCR2 is an emerging strategy to enhance the action of existing and novel immunotherapies. 

In this review, we presented several promising pre-clinical studies that show benefit of combining CXCR2 antagonists with immunotherapy. Much of the current literature is focused on immune checkpoint inhibitors while data concerning CXCR2 antagonism with immunotherapies other than checkpoint blockade are limited and necessitates further study. As CXCR2 antagonists are developed clinically, a challenge will be to determine which patients will benefit from a combination of immune checkpoint inhibitors and CXCR2 inhibition. Due to the associations between high MDSC infiltrates and therapy resistance across cancer types [7], tumors with high levels of MDSCs or patients who have developed resistance to prior lines of therapy may benefit from treatment with a CXCR2 antagonist. However, inconsistencies in MDSC nomenclature in pre-clinical studies remain an obstacle to identifying markers of response that translate clinically. Although, the recent identification of CD14 as a marker of to distinguish G-MDSCs from neutrophils in tumor-bearing mice [20] is a promising step toward overcoming this obstacle. Furthermore, given the role of CXCR2 in maintaining the metastatic niche, CXCR2 antagonists may be particularly useful in patients with late-stage or metastatic disease. However, as immunotherapy moves toward approval for earlier-stage cancers, the potential benefit of adding a CXCR2 antagonist to improve responses warrants investigation. 

Pre-clinical studies targeting other signaling axes involved in MDSC recruitment such as CSF1R and CCR2 showed a benefit of adding CXCR2 inhibition to further deplete MDSC populations and relieve immunosuppression [88,89,90,91,92,93]. Targeting multiple axes of MDSC recruitment in conjunction with immunotherapy may provide a durable anti-tumor response, but toxicities of such triple combinations will need to be carefully considered before advancing this strategy clinically. Early clinical studies with CXCR2 antagonists demonstrated safety and tolerability but failed to meet efficacy endpoints when combined with chemotherapy [71]. Furthermore, blocking CXCR2 may have deleterious effects on tumor-associated neutrophils and the effects of such therapies in relation to the relative amount of anti-tumor N1 to pro-tumor N2 TANs in the TME necessitates further study. Overall, targeting MDSC recruitment by blocking CXCR2 is an emerging strategy with the potential to synergize with existing and novel immunotherapies to improve patient responses and counteract immunosuppressive resistance mechanisms. 

## Figures and Tables

**Figure 1 cancers-13-06293-f001:**
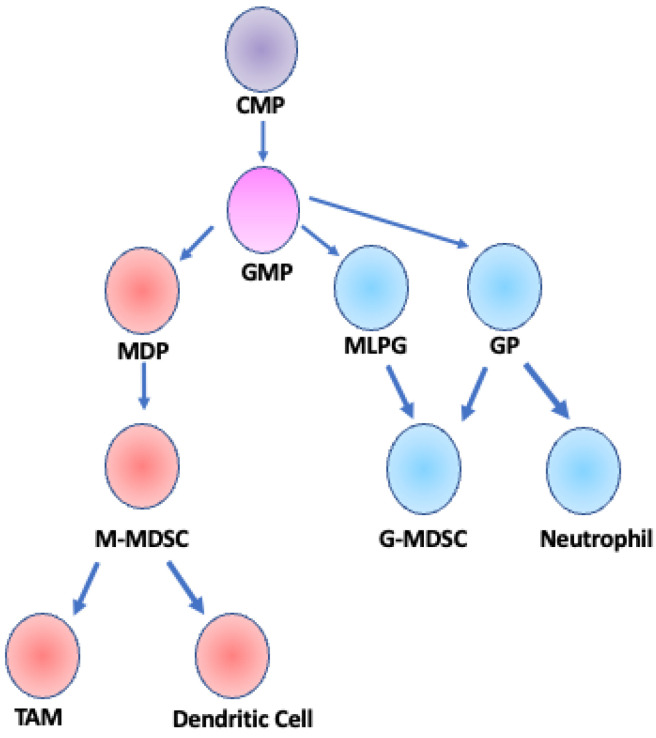
Developmental lineage of MDSCs. M-MDSCs and G-MDSCs arise from a common myeloid progenitor (CMP) and a granulocyte/monocyte progenitor (GMP). GMP cells differentiate into one of three cell types: monocyte-dendritic cell progenitor (MDP), granulocyte progenitor (GP), or monocyte-like precursor of granulocytes (MLPG) [18,19]. M-MDSCs are derived from MDPs and can further differentiate into TAMs or dendritic cells [25]. G-MDSCs can be derived from either MLPGs or GPs and are considered a fully differentiated cell type, distinct from neutrophils that are also derived from GP cells [30]. Adapted from Figure 1 Gabrilovich et al. 2012 *Nat. Rev. Immunol*.

**Figure 2 cancers-13-06293-f002:**
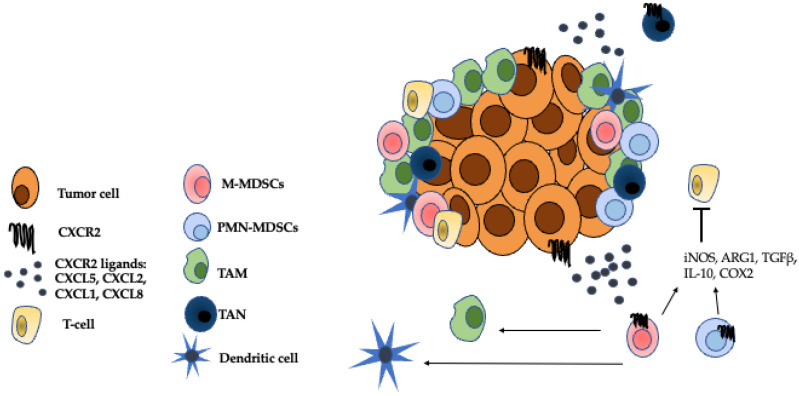
Immunosuppressive cells in the TME. The CXCR2 ligands, CXCL5, CXCL2, CXCL1, and CXCL8 attract G-MDSCs, M-MDSCs, TANs to the TME where they establish an immunosuppressive niche [38]. M-MDSCs can further differentiate into TAMs and inflammatory dendritic cells [25]. Both M-MDSCs and G-MDSCs inhibit the effector functions of T cells through mechanisms such as the secretion of iNOS, ARG1, TGFβ, IL-10, and COX2 [12]. TANS can either have anti-tumor, N1 properties or pro-tumor, N2 properties [23]. The role of TANs in the TME is a source of ongoing research. Adapted from Figure 1 Raman et al. 2007 *Cancer Letters*.

**Table 1 cancers-13-06293-t001:** Ongoing oncology clinical trials targeting CXCR2.

Combination	Drug Name	Indication	Phase	Clinical Trial ID
CXCR2i + hormonal therapy	AZD5069+ enzalutamide	Metastatic castration-resistant prostate cancer	I/II	NCT03177187
CXCR1/2i + anti-PD1	SX-682+ nivolumab	Metastatic pancreatic ductal adenocarcinoma	I	NCT04477343
CXCR1/2i + anti-PD1	SX-682 + nivolumab	RAS-mutated, MSS unresectable or metastatic colorectal cancer	Ib/II	NCT04599140
CXCR1/2i + anti-PD1	SX-682+ pembrolizumab	Metastatic melanoma	I	NCT03161431
CXCR2-transduced autologous TILs + IL-2 + chemotherapy	CXCR2-transduced TILs + aldesleukin + cyclophosphamide and fludarabine phosphate	Metastatic melanoma	I/II	NCT01740557
Single-agent CXCR1/2i	SX-682	Myelodysplastic syndromes	I	NCT04245397

Clinical trial information from clinicaltrials.gov (accessed on 1 December 2021).

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
