# Peer review of "Suppressing MDSC Recruitment to the Tumor Microenvironment by Antagonizing CXCR2 to Enhance the Efficacy of Immunotherapy"

_cancers, 2021, doi:10.3390/cancers13246293_

Round 1

Reviewer 1 Report

        The manuscript covers a highly interesting, clinically important topic. This comprehensive, easy-to-read   manuscript is based on a thorough search of current literature and has appropriate references, mostly published within the last 5 years. All necessary topics are discussed and  the most pressing gaps in knowledge are properly identified. The conclusions are coherent and well supported based on the cited literature. The tables and figures enrich the manuscript, allowing for quick and easy understanding of the text.

        However, the manuscript may benefit from:

        1/ a clarification of relationship between PMN-MDSC and G-MDSC , both terms are correctly used interchangeably in the manuscript but may be confusing for the reader if left without any explanation,

        2/ there is some ambiguity regarding the hallmark markers of MDSC subpopulations in the cited publications. In my opinion, it would be beneficial for the scientific community to describe the particular makers of MDSC used in cited the publications, for example by placing particular phenotype of MDSC in brackets. It is of importance because  apart from G-MDSC and M-MDSC, immature-MDSC (i-MDSC) or early-stage (e-MDSC) have been proposed as subsets of MDSC with own phenotypes,

        3/ improvement of the graphics of Fig.1. Some of the descriptions are placed partially in the cells and partially outside of the cell which render it illegible.

Author Response

      Thank you for the careful review of our manuscript and for the thoughtful comments. We made every effort to respond to the following comments and we hope that you find these changes acceptable. 

  1. Clarification of the relationship between PMN-MDSCs and G-MDSCs was added to section 2.1 as was the statement that the two terms can be used interchangeably. Additionally, all uses of PMN-MDSCs after the initial definition were changed to G-MDSCs to improve clarity and uniformity throughout the piece.
  2. Yes, I agree that the ambiguity in markers of MDSCs is a problem of the field. In order to better incorporate this concern in the review a discussion about the challenges in distinguishing mouse G-MDSCs from neutrophils and M-MDSCs from monocytes was added. Additionally, a brief discussion of the lack of consistency in nomenclature was incorporated into the conclusions in section 9 and described as an obstacle preventing the translation of pre-clinical findings to clinical studies.
  3. Fig 1 was edited such that the descriptions are placed out of the cells. The figure was also centered on the page.

Reviewer 2 Report

In this review, the authors argue that the presence of MDSCs in the tumor immune microenvironment is a major factor associated with nonresponsiveness to tumor immunotherapy and that CXCR2, a chemokine receptor involved in MDSC migration, is a potential therapeutic target. This paper discusses the issue in this light and attempts to present the latest findings on CXCR2-targeted therapy. This is a good review with a careful discussion. The reviewer's comments on this paper are as follows.

1. There are some inaccuracies and inconsistencies in terms and abbreviations. The authors should reorganize them.
- MDSCs
- CXCR2
- TAMs
- TME
- GPCR
- CAR-T

2. Regarding the section 4 "Expression of CXCR2 and CXCR2 ligands is associated with poor response to therapy," the authors should discuss colorectal cancers and DLBCLs because these cancers have also been well studied with CXCR2, MDSCs, and immune checkpoint inhibitors.

3. Regarding the section 6 "Novel CXCR2 based immunotherapy strategies" and the section 8 "Additional strategies to target MDSC recruitment," the authors should further develop the discussion on the possibility of more advanced treatments. The current description seems a bit too conservative for a future outlook.

Author Response

Thank you for the careful review of our manuscript and for the thoughtful comments. We made every effort to respond to the following comments and we hope that you find these changes acceptable. 

  1. The inaccuracies and inconsistencies in the abbreviations were corrected. The terms were defined on their first use and then the abbreviations were used going forward. As an example, the definition of CXCR2 was changed from “C-X-C motif chemokine receptor 2” to “ C-X-C chemokine receptor 2.”
  2. Thank you for pointing out that CXCR2, MDSCs, and immune checkpoint inhibitors have been well studied in DLBCLs and colorectal cancers. References were added to fill this gap. Additionally, a clinical trial in colorectal cancer was added to table 1 (NCT04599140).
  3. The discussions in sections 6 and 8 were further developed to expand on the possibilities of novel CXCR2 based immunotherapy strategies and additional strategies to target MDSC recruitment. Descriptions of future outlooks for these advanced treatments were primarily expanded upon in the section 9 conclusions and future directions.

Reviewer 3 Report

The review by Bullock and Richmond is an interesting perspective on the applications of CXCR2 antagonists to control innate immune cell recruitment into tumors. Comments below are provided to help with clarity and flow.

  1. Several sections of the review read like a list of published literature. The review as a whole would benefit from more conclusions and discussion of the accumulated findings.
  2. Some statements are repeated several times in text. For instance: “blocking CXCR2 as a strategy to enhance response to existing and novel immunotherapies” is repeated 3 times, twice in the last paragraph.
  3. Abbreviations should be defined at first use and used consistently. The review would benefit from proof-reading and more consistent capitalisation, especially in figures and figure legends.
  1. General comment on Figures: there typically are lots of (empty) spaces, which seems not optimally used.
  1. In Section 2.1, discussion of the differences between G-MDSCs and neutrophils is oversimplified, likewise for TAMs and M-MDSCs. There is little discussion of the differences between mouse and human MDSCs.
  2. Neutrophil N1 and N2 phenotypes should be introduced much earlier and discussed in relation to MDSCs.
  3. Line 79 “CD11b+Ly6G+Ly6Glo” should be “CD11b+Ly6G+Ly6Clo”
  4. This statement overlooks many other functions: “PMN-MDSCs primarily act by suppressing CD8+ T-cell 96”
  5. Figure 1 is squashed and awkwardly positioned on the page.
  6. In Section 3, it would make more sense to first explain which immune subsets express CXCR2 in physiological settings and then move on to MDSCs and cancer. Also need to explain which receptors and ligands are expressed in mice and which ones are only in humans. In general, differences between mice and humans should be discussed throughout.
  7. Figure 2 layout could be improved to make better use of available space. Capitalisation is not used consistently and no references are provided for either one of the figures and legends
  8. CXCL8 is defined in the following statement after it has already been discussed in this and previous sections. No mention of the differences between mouse and human. “Ovarian cancer 143 patients with elevated serum levels of the CXCR2 ligands CXCL1 and CXCL2 have in-144 creased intra-tumoral MDSC infiltration and reduced overall survival [34]. Elevated se-145 rum CXCL8 was associated with metastasis and reduced overall survival in pediatric pa-146 tients with rhabdomyosarcoma [35]. CXCL8, also known as IL-8, is a CXCR2 ligand found 147 to be associated with poor response to immunotherapy in particular [36].”
  9. Since this result appears to contradict other findings, some discussion of the differences is warranted: “Interestingly, high expression of CXCR2 was as-154 sociated with longer relapse-free survival in a cohort of triple negative breast cancer pa-155 tients treated with adjuvant chemotherapy [39].”
  10. Please clairfy how this drives colonisation in the bone since experiments are in cell culture: “In studies using ex vivo bone 160 cultures co-cultured with PyMT tumor cells, CXCR2 drove tumor cell colonization in the 161 bone, one of the most common sites of breast cancer metastasis [40].”
  11. This statement requires references: “However, several other studies identified 156 CXCR2 as a key player in establishing the metastatic niche of breast cancer, emphasizing 157 the cell type specificity of CXCR2 expression when considering correlations with clinical 158 outcomes.”
  12. Relevance of this statement to CXCR2 should be clarified: “In a different 201 study using the MOC1 tumor model, depletion of granulocytic MDSCs with a Ly6G+ mAb 202 in combination with anti-CTLA4 led to tumor rejection in 11/11 mice, whereas anti-CTLA4 203 treatment led to tumor rejection in 5/11 mice [48].”
  13. Section 5.1 jumps between granulocytes and MDSCs with no explanation and should be restructured and clarified.
  14. In Section 5.2 the link between obesity and CXCR2 is very confusing and needs to be clarified.
  15. It would be great if the authors could provide a comment on the circumstances and markers that they would use to decide whether CXCR2 inhibition would be beneficial.
  16. What does this statement mean? What functions of CXCR2 do the authors consider to be unnatural? “hijacking the natural functions of CXCR2 is 245 another.”
  17. Statement needs reference: “The production of CXCR2 ligands in the TME drives the recruitment of CXCR2 246 expressing cells into the tumor.”
  18. This sentence contains errors and is confusing: “CXCR2-modifited CAR T-cells 252 showed enhanced intratumoral T-cell infiltration and significantly reduced tumor burden 253 in a murine model of hepatocellular carcinoma that relies on the high expression of 254 CXCR2 ligands by the tumor [53].”
  19. Most of Section 6 reads like a list of literature with little discussion.
  20. Not sure what this sentence means. Please clarify how increasing CXCR2 expression is an alternative to blocking CXCR2 signaling? Are these not completely different situations? One approach increases effector cells whereas the other blocks suppressor cells: “Driving effector cells such as NK cells and T-cells to express 263 CXCR2 is a novel strategy for increasing infiltration of anti-tumor immune cells to the 264 TME and represents an alternative to blocking CXCR2 signaling.”
  21. MK7123 (aka SCH537123) is not explained. It would be good to have a table summarising CXCR2 antagonists.
  22. Third paragraph in Section 7 seems completely out of place in this section as it does not discuss the clinical status of CXCR2 antagonists.
  23. Since CXCR2 is expressed on multiple cell types, would CXCR2 antagonists have an effect on these cells too? This is not discussed.
  24. Last paragraph in Section 7 is very confusing.
  25. In Section 8, the function of CSFR-1 is not defined. A discussion of the opposing effects described in the first paragraph would make this section more meaningful.
  26. In Section 9, perhaps the authors could comment on what criteria could be used to decide whether CXCR2 therapy would be beneficial.
  27. Author contributions are not defined.

Author Response

Thank you for your careful review of our manuscript and for your insightful comments. We made every effort to respond to the following comments and we hope that you find these changes acceptable. 

  1. Discussion of findings were more heavily incorporated in the text to help the flow of the review and make it seem less like a list of published literature.
  2. Efforts were made to edit repetitive statements and introduce more variety in sentence structure in order to make the piece easier to read.
  3. The capitalization in the figures and figure legends was edited to ensure consistency. The manuscript was edited so that abbreviations (TAMs, MDSCs, TME, TANs, etc.) were used consistently after being defined during the first use.
  4. The spacing in figures 1 and 2 were adjusted to minimize white spaces. Figure 1 was centered on the page. In figure 2, the legend and diagram were shifted to minimize blank spaces.
  5. A more thorough discussion of G-MDSCs, neutrophils, TAMs, and MDSCs was incorporated into section 2.1. A discussion of the differences between mouse and human MDSCs and the challenges in identifying surface markers in mouse models was also included.
  6. The introduction of tumor associated neutrophils including N1 and N2 TANs was moved to section 2.1 MDSC Classifications to address the concern of these cell types not being introduced early enough. The more thorough discussion of the potential effects of CXCR2 inhibition on TANs remains in section 7 as part of the discussion on the effects of CXCR2 inhibition on cell types other than MDSCs.
  7. The typo “CD11b+Ly6G+Ly6Glo was corrected to “CD11b+Ly6G+Ly6Clo
  8. The statement, “PMN-MDSCs primarily act by suppressing CD8+ T-cells,” was removed as this oversimplifies the functions of PMN-MDSCs.
  9. Figure 1 was re-positioned in the center of the page and the descriptions were moved below the depictions of the cells to make them easier to read.
  10. In section three, the paragraph was edited to begin with a brief explanation of the role of CXCR2 in a physiological setting before discussing the role of CXCR2 in cancer. An explanation of the ligands and receptors expressed in mice versus humans was also added.
  11. The layout of figure 2 was adjusted to minimize blank space. Capitalizations in the figure and figure legend were edited to be consistent. The proper references were added to the captions for both figure 1 and figure 2.
  12. The wording of this section was adjusted to remove repetitive definitions of CXCL8. An explanation of the difference between mouse and human CXCR2 ligands was also included.
  13. Since the finding that CXCR2 expression correlated with longer relapse-free survival in a cohort of TNBC patients contradicts the other findings presented in this paragraph, a longer discussion of the possibilities for this discrepancy were included. Information on the cell type specificity of CXCR2 expression in the study (neutrophils) was included as well as a comment on how the presence of CXCR2+ neutrophils could lead to better patient outcomes (N1 TANs).
  14. This statement was re-worded to clarify that the experiments were performed in ex vivo cultures of mouse bones and the studies showed that tumor cells were able to colonize these bone explants. While these experiments were not performed in vivo, they provide evidence that CXCR2 plays an important role in the mechanism of tumor cells colonizing the bone. This is relevant because the bone is one of the most frequent sites of breast cancer metastasis.
  15. The following references were added to support the statement that, “several other studies identified CXCR2 as a key player in establishing the metastatic niche of breast cancer.”
    1. Zhu et al.  “CXCR2 + MDSCs promote breast cancer progression by inducing EMT and activated T cell exhaustion,” 2017 Oncotarget
    2. Sharma et al “Host Cxcr2-dependent regulation of mammary tumor growth and metastasis,” 2015 Clinical and Experimental Metastasis
    3. Romero-Moreno et al., “The CXCL5/CXCR2 axis is sufficient to promote breast cancer colonization during bone metastasis,” 2019 Nature Communications
  16. While CXCR2 was not inhibited in the referenced study, this information was included to support the claim that strategies that affect MDSCs – such as CXCR2 inhibition - can enhance the action of immunotherapy. The sentence was reworded to clarify this point, “In a different study using the MOC1 tumor model, depletion of G-MDSCs with a Ly6G+ mAb in combination with anti-CTLA4 led to tumor rejection in 11/11 mice, whereas anti-CTLA4 treatment led to tumor rejection in 5/11 mice [56], providing further evidence for the efficacy of strategies that target MDSCs to improve immunotherapy responses.”
  17. In section 5.1 the terms granulocytic MDSCs and PMN-MDSCs were used interchangeably without explanation. The terms G-MDSCs and PMN-MDSCs refer to the same cell type and clarification was added to section 1 MDSC classifications to explain that these terms are used interchangeably in the literature. In section 5.1, all instances were changed to G-MDSCs to improved clarity.
  18. The description between the link between obesity and CXCR2 was rewritten to improve clarity. In the referenced study, obese mice but not lean mice benefited from the addition of a CXCR2 inhibitor. Another group showed that the inflammation associated with the obese state causes an increase in CXCR2 ligands. This increase in CXCR2 ligands is suggested as a potential mechanism for why the obese mice responded while the lean mice did not.
  19. A more detailed discussion on the circumstances for which CXCR2 inhibition would be beneficial was added to the section 9 conclusions and future directions.
  20. The unnatural functions of CXCR2 that this statement is referring to is the cell type expression pattern of CXCR2 and how engineering effector immune cells that don’t usually express CXCR2 to express CXCR2 is an emerging immunotherapy strategy. However, the wording of this statement is unclear and the sentence was adjusted to read, “Blocking CXCR2 with small molecule antagonists is one strategy to enhance the efficacy of immunotherapy for clinical benefit; hijacking the functions of CXCR2 by driving effector immune cells to express CXCR2 is another.”   
  21. The following references were added to support the statement that, “The production of CXCR2 ligands in the TME drives the recruitment of CXCR2 expressing cells in the tumor.”
    1. Yang et al., “Targeted deletion of CXCR2 in myeloid cells alters the tumor immune environment to improve antitumor immunity,” 2021 Cancer Immunology Research
    2. Grover et al., “Myeloid-Derived Suppressor Cells: A Propitious Road to Clinic,” 2021 Cancer Discovery
  22. This sentence was restructured to remove errors and improve clarity; “In a murine model of hepatocellular carcinoma, treatment with CXCR2-modified CAR T-cells significantly reduced tumor burden and enhanced intratumoral T-cell infiltration.”
  23. Section 6 initially read more like a list of literature. More discussion and analysis of the presented findings was incorporated to try and make this section flow better.
  24. The wording of this sentence was unclear and the phrase, “alternative to blocking CXCR2 signaling” removed.  As you correctly pointed out, increasing effector cells and blocking suppressor cells are two completely different situations, not alternatives.
  25. Relevant CXCR2 antagonists under investigation in clinical oncology trials are summarized in Table 1. Although a new table explaining other antagonists used in pre-clinical studies was not added, Table 1 was updated to include an additional trial left out in the previous version (NCT 04599140).
  26. The third paragraph of section 7 discusses a clinical concern of using CXCR2 antagonists in cancer treatment. The unintended effects of CXCR2 antagonism on neutrophils may be of particular concern when using these inhibitors to treat cancer because of the effects on both N1 and N2 TANs. The wording in this paragraph was adjusted to improve clarity and the section 7 title heading was changed from, “Clinical status of CXCR2 antagonists” to “Clinical status and concerns of CXCR2 antagonists”.
  27. As you correctly point out, the effects of CXCR2 antagonists on cell types other than MDSCs is a concern with using these inhibitors clinically. Please see section 7, paragraphs 2 and 3 for a discussion on the potential unwanted effects of CXCR2 antagonism on neutrophils. A brief description of the other cell types that could be affected by CXCR2 inhibition was also included.
  28. The last paragraph of section 7 was intended to discuss resistance mechanisms as a clinical concern of using CXCR2 antagonists. In particular, rebound increases in CXCR2 ligands and accompanying inflammation were discussed as a concern with using these inhibitors in combination with immunotherapies. This section was edited to improve clarity and better connect it to the other paragraphs in section 7.
  29. CSF-1R was defined as a signaling axis involved in the recruitment of MDSCs similar to the CXCR2 axis. While a more thorough discussion of the function of CSF-1R similar to the first paragraph on CXCR2 would be more meaningful, the description of CSF-1R was brief in order to introduce an additional pathway being explored to limit MDSCs while maintaining the focus of the discussion on CXCR2.
  30. In section 9, the discussion on criteria that could be used to decide whether CXCR2 therapy would be beneficial was expanded. “Given the role of CXCR2 in maintaining the metastatic niche, CXCR2 antagonists may be particularly useful in patients with late-stage or metastatic disease. However, as immunotherapy moves toward approval for earlier stage cancers, the potential benefit of adding a CXCR2 antagonist to improve responses warrants investigation. Given the associations between high MDSC infiltrates and therapy resistance across cancer types [7], tumor with high levels of MDSCs or patients who have developed resistance to prior lines of therapy may benefit from treatment with a CXCR2 antagonist.”
  31. Author contributions were added.

Round 2

Reviewer 2 Report

The authors have addressed this reviewer's comments appropriately.